# Phylogenetic analysis of metastatic progression in breast cancer using somatic mutations and copy number aberrations

David Brown[1],*, Dominiek Smeets[2,3],*, Borbála Székely[4],*, Denis Larsimont[5], A Marcell Szász[4],
Pierre-Yves Adnet[1], Françoise Rothé[1], Ghizlane Rouas[1], Zsófia I. Nagy[4], Zsófia Faragó[4], Anna-Mária Tőkés[4,6],
Magdolna Dank[7], Gyöngyvér Szentmártoni[7], Nóra Udvarhelyi[8], Gabriele Zoppoli[9], Lajos Pusztai[10],
Martine Piccart[11], Janina Kulka[4], Diether Lambrechts[2,3], Christos Sotiriou[1],** & Christine Desmedt[1],**

Several studies using genome-wide molecular techniques have reported various degrees of genetic heterogeneity between primary tumours and their distant metastases. However, it has been difficult to discern patterns of dissemination owing to the limited number of patients and available metastases. Here, we use phylogenetic techniques on data generated using whole-exome sequencing and copy number profiling of primary and multiple-matched metastatic tumours from ten autopsied patients to infer the evolutionary history of breast cancer progression. We observed two modes of disease progression. In some patients, all distant metastases cluster on a branch separate from their primary lesion. Clonal frequency analyses of somatic mutations show that the metastases have a monoclonal origin and descend from a common 'metastatic precursor'. Alternatively, multiple metastatic lesions are seeded from different clones present within the primary tumour. We further show that a metastasis can be horizontally cross-seeded. These findings provide insights into breast cancer dissemination.

[1] Breast Cancer Translational Research Laboratory, Institut Jules Bordet, Université Libre de Bruxelles, Bld de Waterloo 121, 1000 Brussels, Belgium. [2] Laboratory of Translational Genetics, Vesalius Research Center, VIB, Campus Gasthuisberg, O&N IV Herestraat 49, 3000 Leuven, Belgium. [3] Laboratory of Translational Genetics, Department of Oncology, Katholieke Universiteit Leuven, O&N IV Herestraat 49, 3000 Leuven, Belgium. [4] Second Department of Pathology, Semmelweis University, Üllői út 93, 1091 Budapest, Hungary. [5] Department of Pathology, Institut Jules Bordet, Bld de Waterloo 121, 1000 Brussels, Belgium. [6] 2nd Department of Pathology, MTA-SE Tumor Progression Research Group, Semmelweis University, Üllői út 93, 1091 Budapest, Hungary. [7] Semmelweis University Cancer Center, Semmelweis University, Tömő u. 25-29, 1083 Budapest, Hungary. [8] Surgical and Molecular Tumor Pathology Centre, National Institute of Oncology, Ráth György u. 7-9, 1122 Budapest, Hungary. [9] University of Genova and Istituto di Cura a Carattere Clinico e Scientifico Azienda Ospedaliera Universitaria San Martino—Instituto Nazionale Tumori, Largo Rosanna Benzi 10, 16132 Genoa, Italy. [10] Yale University, Cedar Street 333, New Haven, Connecticut 05620, USA. [11] Department of Medical Oncology, Institut Jules Bordet, Université Libre de Bruxelles, Bld de Waterloo 121, 1000 Brussels, Belgium. * These authors contributed equally to this work. ** These authors jointly supervised this work. Correspondence and requests for materials should be addressed to C.S. (email: christos.sotiriou@bordet.be) or to C.D. (email: christine.desmedt@bordet.be).

Cancer-related mortality is almost always due to metastatic dissemination of the primary disease. While research continues to unravel the molecular underpinnings of the metastatic cascade, it is increasingly recognized that profiling of advanced disease could help elucidate such biological phenomena as distant recurrence and the emergence of *de novo* resistance to therapy.

A handful of studies using genome-wide molecular techniques have begun to explore the clonal relationships between primary and matched metastatic tumours in diverse types of neoplasia including pancreatic[1,2], clear-cell renal cell[3], high-grade serous ovarian[4–6] and prostate cancer[7,8]. Despite the small cohort sizes and, too often, a limited number of matched metastases for each patient, these pioneering efforts brought forth thought-provoking findings such as the first quantitative model of cancer progression from onset of the founder mutation to metastatic dissemination[2], the occurrence of organ specific lineages[1], monoclonal[3–8], as well as its counterpart, polyclonal seeding[7,8], horizontal cross-seeding between distant metastases[6,8], and finally homing of metastatic cells to the primary tumour bed[7].

While yet other studies continue to highlight the potential of genomic analyses from small cohort sizes to decipher the origins of intra-tumour heterogeneity and its contribution to metastatic dissemination[9,10], in-depth knowledge is currently lacking for breast cancer. Several studies have tackled this issue[11–19]. However, while early attempts were constrained by the development of high throughput genomic techniques, more recent endeavours were, on the other hand, limited in scope by the availability of multiple-matched metastases. Noteworthy exceptions are the work of Juric *et al.*[15] and Murtaza *et al.*[16], both *n*-of-1 fast autopsy studies, where the authors uncovered the mechanisms of resistance to a PI3K-inhibitor and lapatinib, respectively. Despite this, it remains difficult to discern any pattern of metastatic progression due to the small number of patients.

To further investigate breast cancer progression, we applied phylogenetic techniques on data generated using whole-exome sequencing, custom ultra-deep resequencing and copy number profiling. The primary tumours and their associated metastases were obtained from ten autopsied patients. We observed two modes of metastatic progression. In the majority of cases, all distant metastases cluster on a branch separate from their primary lesion. Clonal frequency analyses of somatic mutations show that the metastases have a monoclonal origin and descend from a common 'metastatic precursor'. Alternatively, the primary tumour is clustered alongside metastases with early branches leading to distant organs. Finally, we show that a distant metastasis can be horizontally cross-seeded confirming previous results observed in other types of neoplastic disorders[6,8] and lending further support to the self-seeding hypothesis[20].

## Results

**Characteristics of patients and samples**. We reviewed the database of the institutional autopsy programme of the second Department of Pathology at Semmelweis University. From 50 deceased metastatic breast cancer patients, whose corpses underwent autopsy between 2001 and 2012, ten patients for whom >1 μg double-stranded DNA from the primary breast tumour, a non-cancerous tissue as germline reference, and at least one metastatic sample was available, were selected. Eight patients were diagnosed with early stage disease among whom, one was diagnosed with a single liver metastasis (5/87). Three patients (3/92, 5/87 and 6/91) received neoadjuvant chemotherapy before surgery while the remaining five patients (4/71, 7/67, 8/82, 9/68 and 10/80) were treated with breast surgery followed by adjuvant systemic therapy according to standard of care. The remaining two patients (1/69 and 2/57) were diagnosed with *de novo* metastatic disease and deceased before receiving any systemic or surgical treatment. The patient clinico-pathological characteristics are provided in Supplementary Data 1 while the clinical history and autopsy findings are detailed in Supplementary Notes 1–10 corresponding to patient 1/69 to 10/80. The lesions profiled are described in Supplementary Data 2. All samples from the *de novo* metastatic patients were collected post-mortem while, for the remaining patients, the primary tumours were collected at surgery and the distant metastases, in addition to one case of local recurrence, were collected at autopsy. On average, three distant metastatic lesions were profiled per patient.

**Indexing of somatic mutations and copy number aberrations**. We used whole-exome sequencing to index somatic mutations from 51 samples (median coverage $40 \pm 18 \times$) followed by orthogonal validation using Sequenom MassARRAY to exclude false positive calls and targeted amplicon ultra-deep sequencing (median coverage $11,390 \pm 5,646 \times$) to obtain accurate variant allele frequencies (VAFs). The list of single nucleotide variants (SNVs) from each patient is provided in Supplementary Data 3. We supplemented this with high density single nucleotide polymorphism (SNP) arrays to characterize the underlying copy number aberrations (CNAs) in 64 matched samples (Supplementary Data 4). We further devised a multiple tier system to ascribe a confidence level to each indexed mutation. Between 27 and 305 non-synonymous SNVs per patient were successfully validated up to tier-3 level and after applying defined quality criteria, a total of 56 samples with either CNA or sequencing data remained for downstream analysis.

**Phylogenetic reconstruction of metastatic progression**. Metastases are clonally related and originate from cells disseminated at various stages of the disease. Thus, they inherit varying fractions of genomic alterations from their parental lineage, followed by acquisition of private alterations. Provided the genomic alterations under investigation are fully clonal, phylogenetic inference can be used to investigate lineage tracing of metastases within a patient. Therefore, we used a maximum parsimony criterion to infer the sequence of genomic alterations occurring during metastatic progression. Figure 1a–f illustrates the results obtained in patient 2/57. Whenever the two phylogenies obtained from SNVs and CNAs were consistent, these were graphically represented as a combined tree. In the case of SNV-based phylogenies containing unresolved nodes, so called soft polytomies, we used the corresponding tree generated from CNAs as the correct phylogeny on account of the greater number of aberrations from SNP arrays, allowing for a unique solution to tree reconstruction.

The combined use of SNVs and CNAs demonstrated the presence of reversions, that is, SNVs predicted as present in a sample from the ancestral state reconstruction but were not detected in the particular sample. For the predicted reversions, we excluded the possibility that these were due to false negative calls attributable to inadequate sequencing coverage depth or the occurrence of the mutations at subclonal cell frequencies based on power calculations described in Carter *et al.*[21] (Supplementary Fig. 1). Instead, Fig. 2a–i shows two clusters of reversions in the metastasis to the pylorus from patient 8/82, which can be attributed to loss of heterozygosity at chromosome 1p and 17p in that lesion. We further encountered a similar phenomenon in patient 9/68 (Supplementary Fig. 2) where the

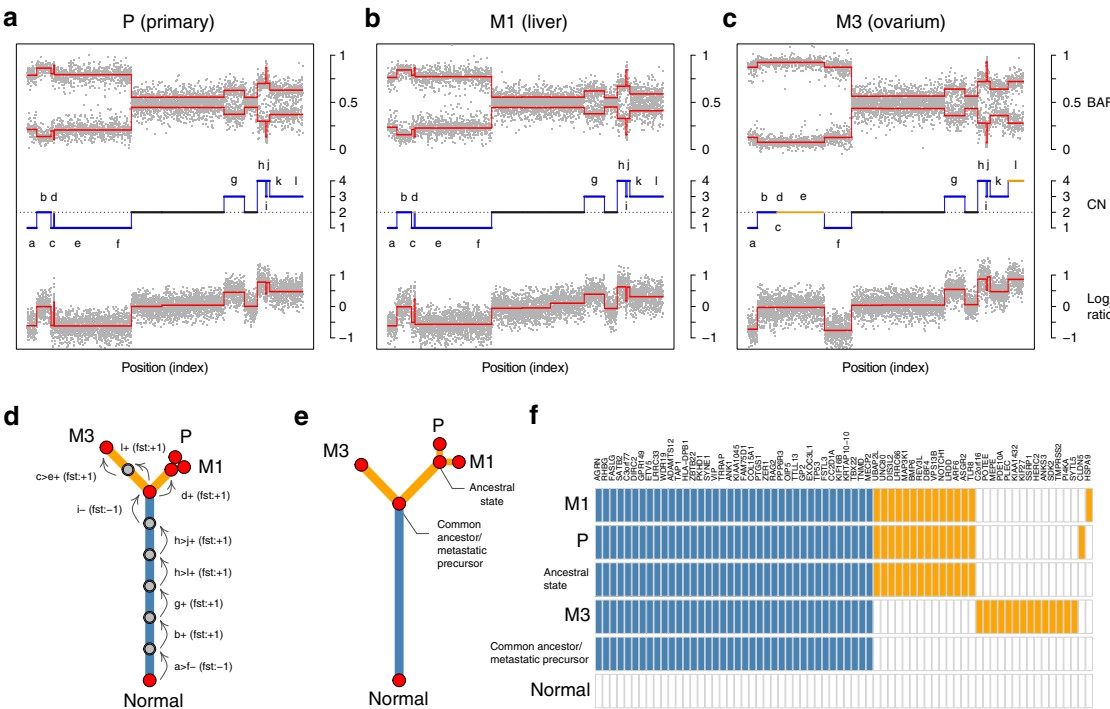

**Figure 1 | Phylogenetic inference from SNVs and CNAs.** Fully clonal somatic CNAs of chromosome 3 for (**a**) the primary tumour, (**b**) the liver and (**c**) the ovarian metastases of patient 2/57. The tracks are in descending order, the B Allele Frequency (BAF), integer copy number (CN) state and log₂ ratios. The phylogenetic reconstruction is displayed in **d**, where arrows indicate aberrations with annotations besides detailing coordinates and event type. Convergent evolution is exemplified by the focal amplification of region d in **a,b** but a broad gain of c–e in **c**, in all three cases leading to copy-neutral loss of heterozygosity of region d. The CNAs are colour coded with 'early' events in blue, 'late' events in orange, and diploid regions not contributing to the phylogenetic tree in black. (**e**) Concordant phylogeny obtained from tier-3 SNVs and (**f**) ancestral state reconstruction for the same samples. The scale bars in (**d,e**) represent one CNA and 10 SNVs, respectively.

change in copy number status of chromosome 19p can be tracked across the phylogenetic tree and explains the reversion of the mutation in *F2RL3*. The occurrence of reversions has seldom been acknowledged in the literature and despite growing interest in the inference of phylogenies from single or multiple samples, several approaches have falsely relied on the hypothesis of 'no back mutation'. Therefore, these examples serve as cautionary tale when performing such analysis without properly matched sequencing and CNA data.

**Disease dissemination via a metastatic precursor.** We applied the same workflow to all patients for whom both SNV and CNA data were available. We refer to 'early' and 'late' alterations when occurring in the trunk or the branches of the phylogenetic trees, respectively. A representative combined phylogeny for patient 7/67 is illustrated in Fig. 3. This patient was diagnosed at the age of 54 with an ER-/PgR-/HER2-primary breast cancer. She deceased 3 years later despite surgery and several lines of systemic treatments. All the distant metastases clustered together and descended from what we refer to as the 'metastatic precursor'. We computed the clonal frequencies from the VAFs, the global cancer cell fraction (CCF) and the copy number states for each SNV. These are represented as pairwise comparisons of samples (Fig. 3d). Similarly, the phylogenies of patients with early breast cancer disease (4/71, 8/82, 9/68 and 10/80), and the one from patient 5/87 with a single liver metastasis at initial diagnosis, further confirmed the case of patient 7/67 (Fig. 5a; Supplementary Figs 4 and 5). Distant metastases probably arose via a seeding event to an initial 'metastatic precursor' from the primary tumour and in absence of the latter,

removed at surgery, the source of further dissemination to additional organs occurred by metastasis-to-metastasis disseminations. Our observation suggests that for breast cancer patients diagnosed at an early stage and undergoing curative intent surgery, who represent the majority of patients, cascading disseminations from metastases appears to be a major route of tumour progression.

For seven patients from our cohort, multiple samples from the primary tumour were available (3/92, 4/71, 5/87, 6/91, 8/82, 9/68 and 10/80). In two cases, a particular region of the primary tumour was more genetically related to the distant metastases. For patient 3/92, the phylogenetic tree based on CNAs (Supplementary Fig. 9) show that the primary tumour sample (P3) was clustered alongside the metastases to the liver (M2) and pancreatic lymph node (M3) while for patient 6/91, the tree inferred from tier-3 SNVs (Supplementary Fig. 7) shows that the primary tumour sample (P3) is clustered alongside the metastases to the brain (M1) and liver (M2). Apart from these two exceptions, in all the other patients, the different samples from the primary tumour were clustered together separate from their associated distant metastases (Fig. 2; Supplementary Figs 7 and 9).

**Multiple seeding events from the primary tumour.** A contrasting clinical and biological condition to the dissemination via a 'metastatic precursor' is illustrated by the case of patient 2/57 (Fig. 4). This patient was a *BRCA1* germline mutation carrier diagnosed at the age of 38 with an ER-/PgR-/HER2-metastatic breast cancer. She did not receive any systemic treatment and deceased 1 month after initial diagnosis. Analysis

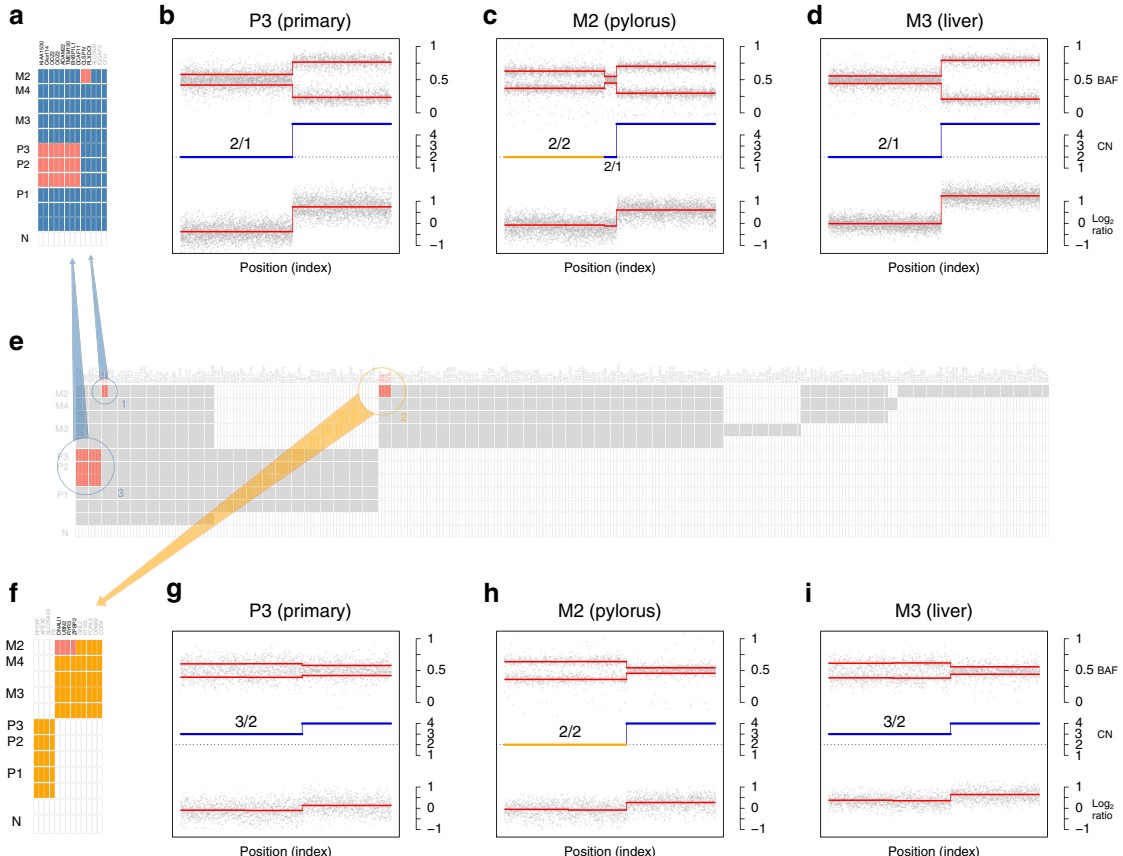

**Figure 2 | Reversions in SNVs are explained by underlying CNAs.** (**a,f**) 'Early' and 'late' tier-3 SNVs, respectively, which were predicted to be reversions in the metastasis to the pylorus of patient 8/82. These were clustered on chromosome 1p and 17p. (**b–d, g–i**) Fully clonal somatic CNAs of chromosome 1 and 17, respectively, for the primary tumour, the pylorus and the liver metastases ordered according to their genomic coordinates. In each panel, the tracks displayed are in descending order, the BAF, the integer based estimation of CN and the log$_2$ ratios. (**e**) Heat map representing the ancestral state reconstruction. The loss of heterozygosity at chromosome arm 1p and 17p in M2 explains the absence of these mutations.

of her primary tumour (P) and two distant metastases revealed two independent seeding events from the primary leading to the ovarian (M3) and to the liver (M1) secondary lesions, respectively (Fig. 1d,e). The phylogenetic reconstruction from the CNA profile of the metastasis to the adrenal gland (M2) revealed that this lesion originated from a precursor shared with the liver metastasis (Fig. 4b). However, the adrenal gland lesion displayed both SNVs acquired 'late' in the evolutionary history of the clade composed of the primary tumour and liver metastasis as well as SNVs private to the ovarian metastasis. Pairwise comparisons of the clonal frequencies of tier-4 SNVs showed that those private to the primary tumour and liver metastasis clade (segment 4) were also present at full clonal frequencies in the adrenal gland metastasis (Fig. 4c) in agreement with the phylogeny inferred from the CNA profiles. The 'late' SNVs private to the ovarian metastasis (segment 2) were observed at subclonal frequencies in the adrenal gland metastasis. We resequenced M2 and obtained similar results (Supplementary Fig. 3). Our results imply that circulating metastatic cells, disseminated by the ovarian metastasis, horizontally cross-seeded the already metastatic adrenal gland and confirm previous observations in ovarian[6] and prostate[8] cancers further lending support to the hypothesis of tumour self-seeding[20].

In the additional advanced stage breast cancer patient (1/69) who was *de novo* metastatic and died in the weeks following her diagnosis without receiving any systemic treatment,

the primary sample was also found clustered alongside distant metastases (Fig. 5b; Supplementary Fig. 6). We observed a similar early seeding to distant organs (primary to pleura in 1/69 and primary to ovarium in 2/57) followed by subsequent late seeding events to additional organs from either the primary lesion (primary to aorta in 1/69 and primary to liver in 2/57) or from already established metastases (mediastinal soft tissue to mediastinal lymph node or vice versa in 1/69 and liver to adrenal gland or vice versa in 2/57).

**Contralateral breast tumours originate from primary tumours.** Two patients from our series were diagnosed with a metachronous contralateral breast tumour. Patient 6/91, 1 year and 10/80, 10 years after initial diagnosis. In patient 6/91, the phylogenetic reconstruction based on tier-3 SNVs showed that the contralateral left tumour (M3) was the earliest branching but shared a substantial fraction of the truncal SNVs (Supplementary Fig. 7). This contrasts with the case of patient 10/80 where the CNA profiles show that the contralateral left tumour (M1) originated from a daughter lesion shared with the liver metastases (M2 and M3) (Supplementary Fig. 9). Nonetheless, both cases confirm the clonal relatedness of the contralateral tumour with the initially diagnosed breast cancer. Together with recent reports[22,23], this calls into question the current practice of considering metachronous contralateral tumours as second primary cancers. Since treatment strategies offered to patients

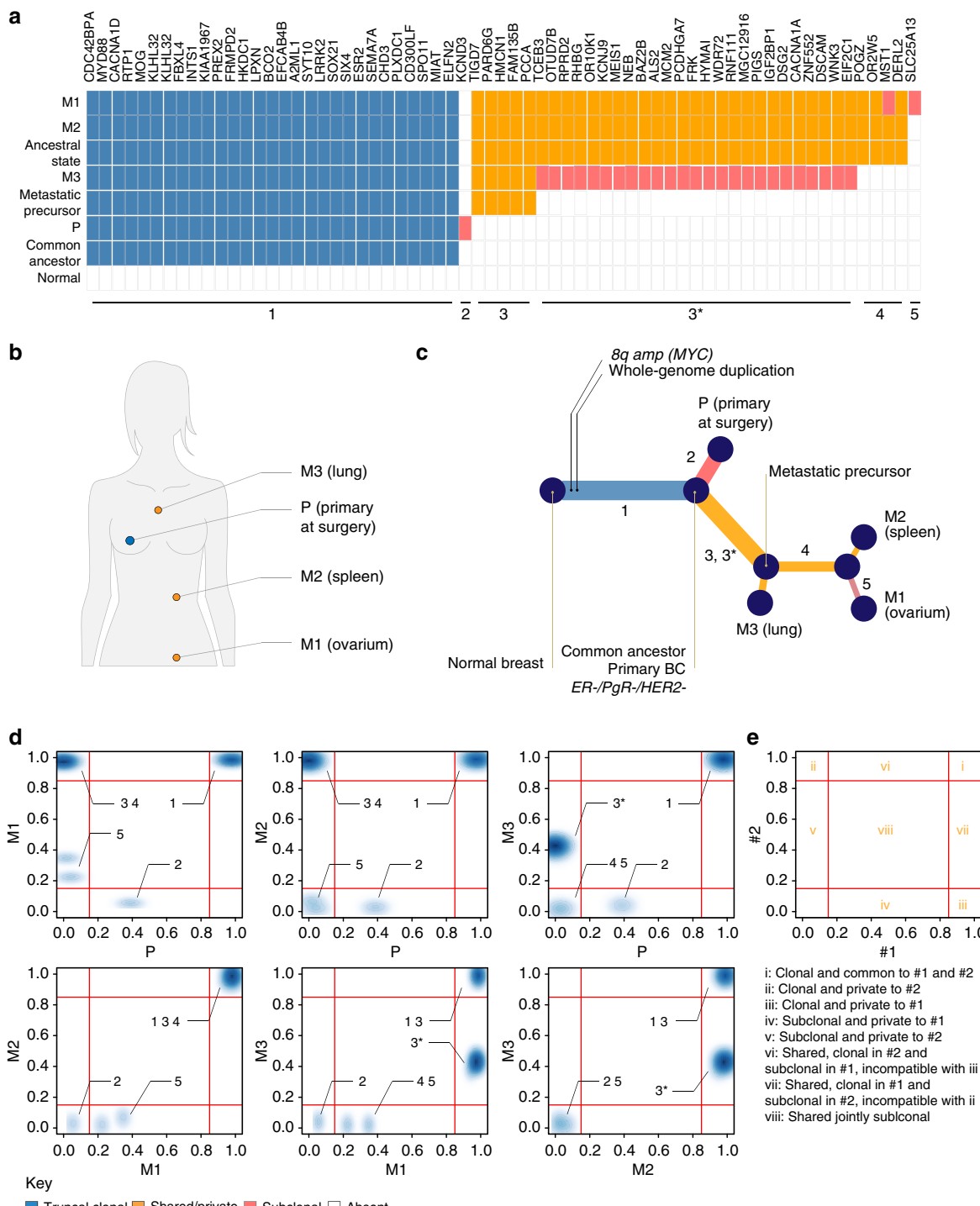

**Figure 3 | Phylogenetic reconstruction of breast cancer progression in patient 7/67.** (**a**) Ancestral state reconstruction of tier-3 SNVs with the anatomic location of the profiled lesions is depicted in **b**. (**c**) Combined phylogenetic tree obtained from CNAs and SNVs and (**d**) pairwise comparisons of clonal frequencies of tier-4 SNVs. The branches of the phylogenetic tree are labelled 1–5 and the location of these mutations in pairwise comparisons is indicated in **d**. (**e**) Schematic representation of the pairwise comparison of two fictitious samples. Mutations in i, ii and iii are fully clonal being either common to the two samples and thus inherited from their parental lineage or private to either one. Mutations in iv and v are private and subclonal to either samples. They are expected to have occurred after the divergence of the two lineages and after mutations located in ii and iii, respectively. Mutations in vi and vii are shared between the two samples but are fully clonal in one and subclonal in the other. If the two samples share a common parental origin, these mutations are incompatible with fully clonal mutations occurring in ii and iii, respectively. A possible scenario explaining their occurrence is that vi and vii are mutually exclusive and that sample #1 seeded #2 giving rise to vi or vice versa for vii. The subclonal frequencies could then be explained by intra-tumour heterogeneity in the tumour mass. Alternatively, mutations in vi and vii could find their origin in horizontal reseeding from a third sample.

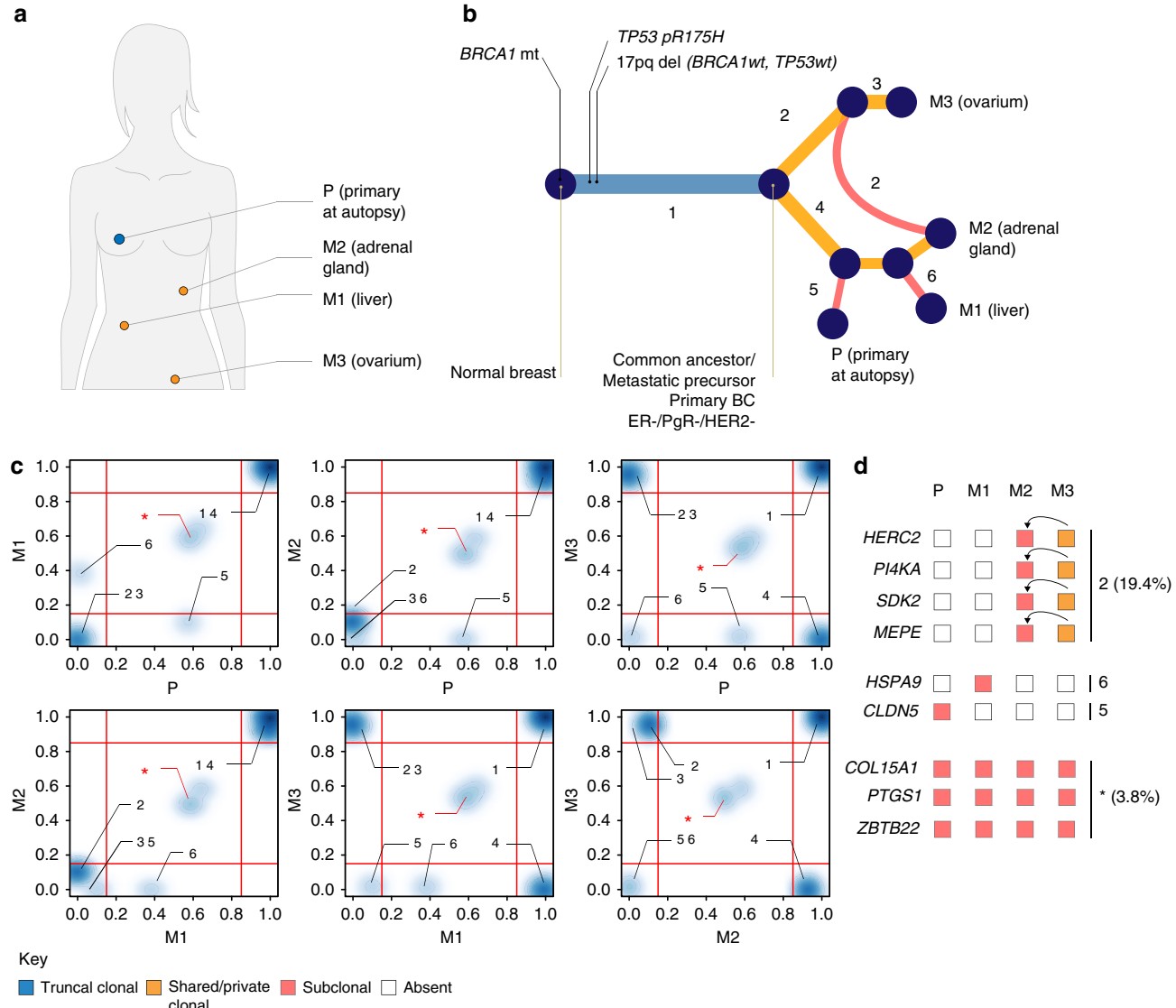

**Figure 4 | Phylogenetic reconstruction of breast cancer progression in patient 2/57.** (**a**) Anatomical representation of tumour lesions profiled, (**b**) combined phylogenetic tree obtained from CNAs and SNVs, and (**c**) pairwise comparisons of clonal frequencies from tier-4 SNVs. The branches of the phylogenetic tree are labelled 1–6 in **b** and the location of these mutations in pairwise comparisons is indicated by the corresponding label in **c**. Mutations in segment 2 at full clonal frequencies in M3 and subclonal frequencies in M2 indicate horizontal seeding, highlighted by the red segment 2 in **b**. A heuristic interpretation of the different possible scenarios is given in **d**. Only three mutations in total were in the unexplained configuration *, one in 5, one in 6 and 12 in configuration 2. Excluding mutations in the configuration *does not influence the topology of the phylogeny. The numbers in parentheses in **d** give the percentage of all tier-4 SNVs.

differ widely between early and advanced stage breast cancers, it is imperative to determine in practice whether contralateral tumours represent a metastatic deposit of the primary tumour.

**Evolution of genomic alterations during cancer progression.** We computed for each lesion, a normalized phylogenetic branch length, which is the ratio of the path from the common ancestor to the given lesion relative to the common trunk (Fig. 6a,b). This represents the extent of genomic alterations that accumulated since the first metastasizing event took place irrespective of the mode of progression. With few exceptions, the pattern observed from tier-3 SNVs mirrors the one from CNAs. In patients 1/69, 2/57 and 4/71, who all died from their disease at most 1 year after initial diagnosis, the bulk of evolutionary changes occurred 'early' in the trunk of the phylogenetic tree. At the other extreme, in patients 8/82, 9/68 and

10/80, who had a longer disease history, the SNV profile is unco-ordinated with the CNA whereby the former shows that most of the evolutionary changes occurred 'late'. Figure 6c,d shows the correlation of the average normalized phylogenetic branch lengths with overall survival. Although the number of patients is small, we observed a positive correlation for both CNAs and SNVs.

In patients 8/82 and 10/80, we observed a high mutational burden which showed evidence of increased C > T substitutions at NpCpG trinucleotides. This pattern of substitution is reminiscent of mutational signature 13 in Alexandrov *et al.*[24] Figure 7a,b shows the pattern of substitutions observed in the trunk, that is, 'early', and in branches, that is, 'late', during the evolutionary cascade of these two patients. Thus, it is possible that, in at least these two patients, the activation of the APOBEC family of cytidine deaminases caused an accumulation of mutations which uncoupled the SNV and CNA profiles.

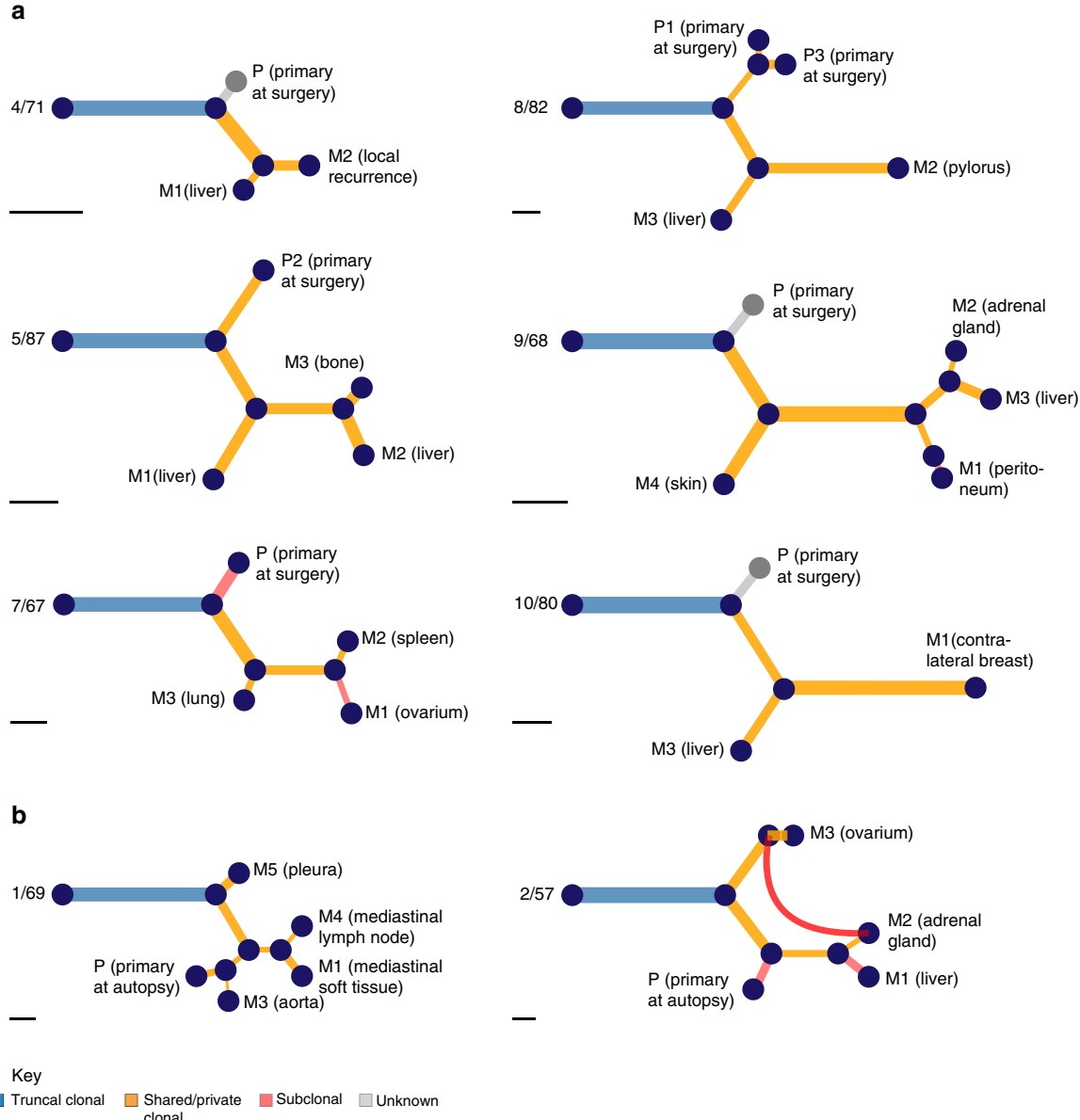

**Figure 5 | Combined phylogenies representing metastatic progression across eight patients.** (**a**) Phylogenies of early stage patients who underwent primary surgery followed by systemic treatment and (**b**) phylogenetic trees obtained from advanced stage treatment naïve and *de novo* metastatic patients. The same colour code as in previous figures is used to depict 'early' and 'late' events. For visual purposes, all the trees were globally rescaled such that the trunks of the trees have the same length. The scale bars at the bottom represent 10 SNVs and provide an indication of the original length of the trees. For patients 4/71 and 9/68, the primary tumour samples removed at surgery were exome sequenced and putative tier-1 somatic mutations were further validated by Sequenom MassARRAY and ultra-deep amplicon sequencing. However, the corresponding SNP arrays showed that these samples had CCFs below the set threshold of 30% for phylogenetic reconstruction. Nonetheless, for these two particular samples, tier-3 SNVs were included in the construction of the phylogenetic trees for SNVs on account that the lesions had been removed several years prior to the diagnosis of distant relapses and autopsy. Similarly, for patient 10/80, the primary tissue samples did not pass the filtering criteria of tier-3 level. The thickness of the branches leading to these nodes is therefore irrelevant. These are displayed in grey.

Patient 10/80 harboured an 'early' amplification of the APOBEC cluster on chromosome 22 (Supplementary Data 4) while patient 8/82 harboured an 'early' *APOBEC3B* D316N mutation (Supplementary Data 3).

## Discussion

Herein, we applied phylogenetic techniques to infer the evolutionary history of breast cancer progression from an autopsy cohort of ten patients. In contrast to previous reports, which compared single metastasis and primary tumour pairs only or multiple-matched metastases and primary tumours for no more than two patients, the availability of a larger number of patients with matched primary and multiple metastatic samples was critical to our study for deciphering the routes of dissemination underlying metastatic progression.

We observed two possible scenarios. The most frequent implied a single successful seeding event from the primary tumour followed by metastasis-to-metastasis cascading disseminations, whereas the second involved multiple seeding events

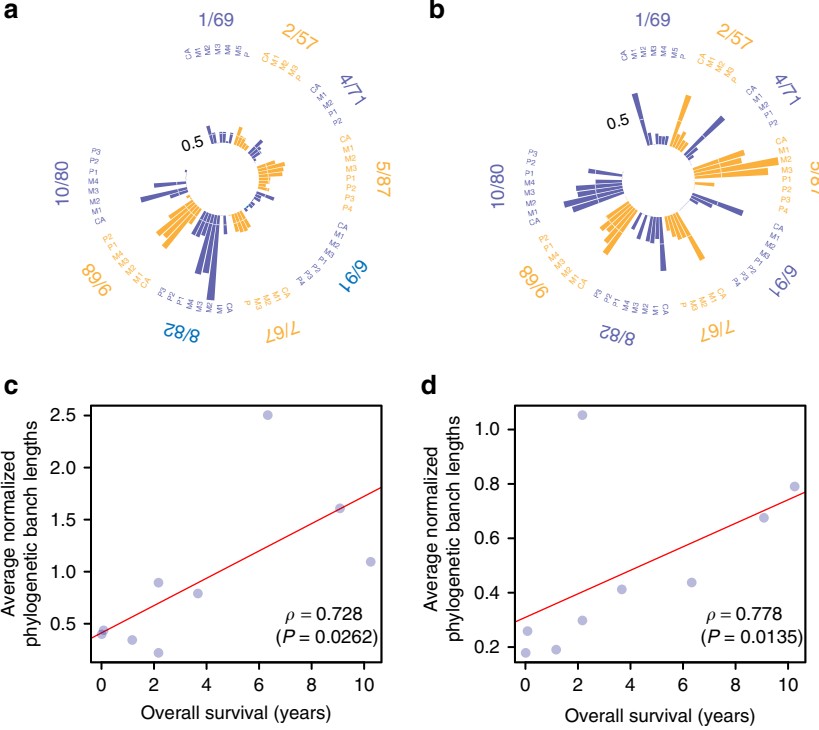

**Figure 6 | Dynamics of genomic alterations during metastatic progression.** Normalized phylogenetic distance for each sample profiled. These are obtained as the ratio of the path from the common ancestor node to the given sample relative to the trunk of the tree for (**a**) SNVs and (**b**) CNAs. (**c**,**d**) Correlation of the average phylogenetic distances with overall survival for SNVs and CNAs, respectively.

from the primary tumour alongside daughter metastasis-to-metastasis disseminations. This dichotomy coincides with the clinical history where, except for patient 5/87, descent from a common metastatic origin was observed in patients diagnosed with early stage breast cancer, whereas multiple seeding events from the primary tumour occurred in patients diagnosed with advanced stage disease.

The role of primary tumour resection in *de novo* metastatic breast cancer patients is unclear, and there is currently no consensus whether this procedure confers a survival benefit. A recent open label trial did not support primary surgery in *de novo* metastatic patients progressing to front-line chemotherapy[25] and in a subgroup analysis of a Turkish study[26], patients with multiple liver and lung metastases did worse in the primary surgery group consistent with earlier reports that surgical excision of the primary tumour might enhance the growth of micrometastases[27–29]. However, in the same trial by Soran *et al.*[26], the authors observed an increased progression free survival for primary tumour resection in ER + /HER2-*de novo* metastatic patients with solitary bone metastases. Thus, our observations suggest that surgical excision of the primary tumour might reduce metastatic dissemination in selected cases hence providing a potential biological rationale for this practice. Similarly, there is no strong recommendation showing overall survival benefit from surgical resection of oligo-metastases in breast cancer. From our analyses, metastatic lesions constitute an additional source of seeding and heterogeneity in advanced breast cancer. Our cohort is too small to derive practice-changing evidence, but supports the concept that resecting isolated metastases may be of clinical benefit in oligo-metastatic breast cancer patients. In both cases, results from larger, prospective studies are warranted.

We reckoned that the number of 'late' SNVs and CNAs, should increase as distant metastases evolve and should give an indication, albeit approximate, of the time elapsed since they last diverged from their common ancestor. Indeed, we observed a positive correlation between overall survival and the average normalized phylogenetic branch lengths. This can be explained by the fact that patients 1/69 and 2/57 were *de novo* metastatic and consistent with those two patients, 4/71 also had a very short distant metastasis free and overall survival. At the other extreme, in patients 8/82, 9/68 and 10/80 who relapsed more than 4 years after initial diagnosis, the extent of 'late' genomic alterations were commensurate with the survival of the patients. These results suggest, not unexpectedly, that metastases from patients with longer cancer histories are genetically more distant from their 'common ancestor' or their primary tissue of origin than those of patients with a shorter cancer history.

Evidence has been accumulating in the literature regarding treatment-induced genomic remodelling[15,16,30–37], especially implicating *ESR1* and *PTEN* alterations in endocrine and PI3K-inhibitor resistance, respectively. In our series, four out of the five ER-positive patients received aromatase inhibitors. However, no *ESR1* mutations have been detected in their distant metastases. None of the patients received any PI3K-inhibitor making it impossible to evaluate resistance mechanisms associated to this treatment.

Overall, by characterizing the genomic alterations that shape metastatic genomes, we have gleaned new insights into the dissemination patterns of breast cancer with potential clinical implications: (1) cascading dissemination from metastases appears to be a major route of metastatic progression in early, radically resected breast cancer and (2) primary tumours at diagnosis may not adequately represent advanced metastatic disease advocating the need for genetic characterization of multiple metastatic lesions. The very recent technical advances in the assessment of circulating tumour DNA[38] may both allow

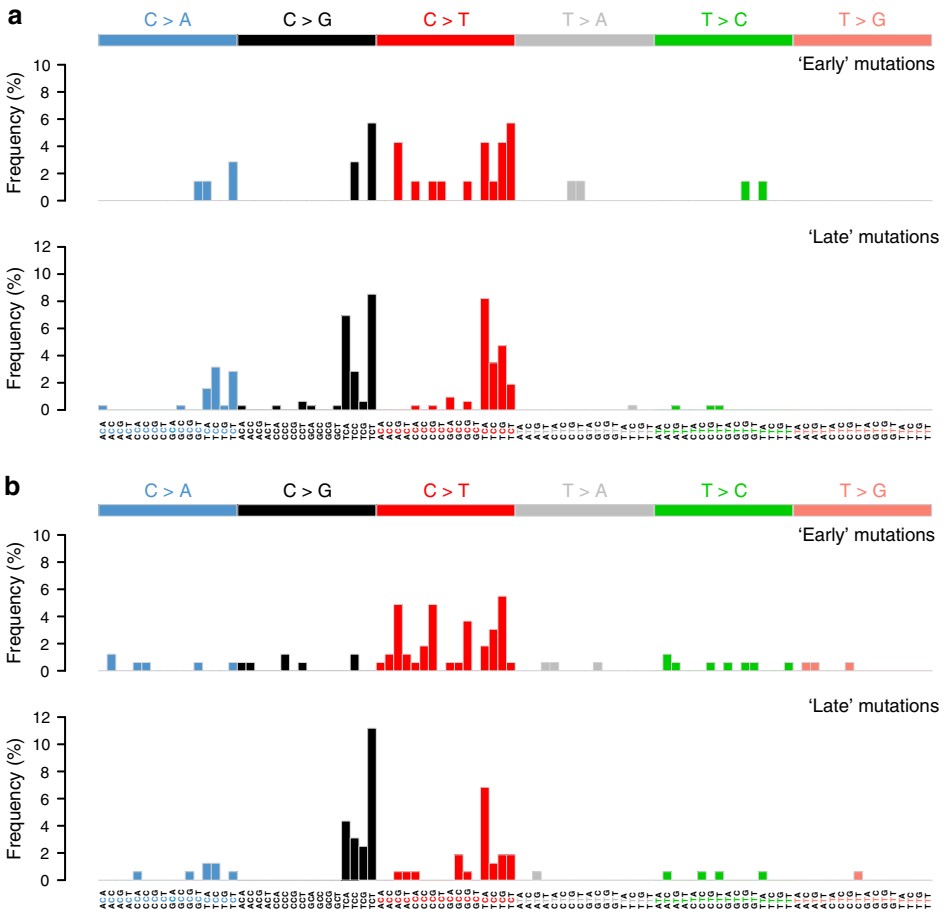

**Figure 7 | Distribution of substitutions during metastatic progression.** Frequency of the different types of substitutions for tier-3 SNVs in patients (**a**) 8/82 and (**b**) 10/80. These are grouped as 'early' and 'late' according to their occurrence in the respective phylogenetic trees.

the early detection of micrometastatic disease before recurrence and may better capture tumour heterogeneity of metastatic disease guiding the best therapeutic options for early and advanced breast cancer patients in the future. Genomic alterations uniquely defining breast cancer metastases from an aetiological standpoint, and therapeutic agents tackling them are yet to be found.

## Methods

**Patients and samples.** The average period between the time of death and acquisition of autopsy samples was 2.8 days (1.5–4.2). Between the time of death and dissection, the cadavers were kept at 4 °C. At autopsy, complete external and detailed internal examinations were performed. The organs were thoroughly examined, weighed and tissue samples were taken. All tissue samples were fixed in formalin and embedded in paraffin as part of routine workup. All bone samples were decalcified using EDTA solution to preserve the antigenicity of the tissue proteins. Hematoxylin-eosin (HE) stained sections were reviewed by J.K., A.M.S. and B.S. to confirm the presence and percentage of invasive carcinoma as well as other tissue composition. A detailed description of the clinico-pathological characteristics of each patient is provided in Supplementary Data 1. This project was approved by the Institutional Review Board (IKEB 185-1/2007). This study is retrospective in nature and part of a larger institutional-based autopsy programme carried out at the Semmelweis University. It did not impact treatment decision for the patients involved and received approval from the ethical committee of the Semmelweis University.

**Pathological characterization of samples.** The stage of primary tumours was reclassified based on the 7th version of TNM classification system. Immunohistochemistry (IHC) and fluorescence *in situ* hybridization (FISH) were performed on 4 μm tissue sections. All the samples underwent centralized IHC for the oestrogen (ER) and progesterone (PgR) receptor status and IHC/FISH characterization for HER2 receptor status. Hormone receptor and HER2 status

were assessed by IHC on all samples with an automated immunostainer system (Ventana Benchmark, Roche Diagnostics, Mannheim, Germany) according to the manufacturer's instructions with the following antibodies: ER–SP1 rabbit monoclonal (Ventana #790-4324) ready to use kit, PgR–1E2 rabbit monoclonal (Ventana #790-2223) ready to use kit, Ki-67–MIB1 (DAKO #M7240, Carpinteria, CA, USA) at dilution 1:100 and HER2/neu–4B5 (Ventana #800-2996) ready to use kit. Hormone receptor status was evaluated using Allred-Quick scoring system by two investigators independently (A.M.S. and B.S.). Ki-67 index was measured as the ratio of the positive tumour cell nuclei in the sample. IHC assessment of Ki-67 was evaluated only on the primary tumours. HER2 positivity was primarily defined at protein level using IHC and supplemented by FISH using Poseidon probes (Kreatech Diagnostics, #KBI-10735, Amsterdam, Netherlands). HER2 IHC was evaluated according to the modified standard protocol that is, positive by IHC only if more than 30% of tumour cells show strong, complete membrane reaction. FISH was performed on samples with IHC 2 + and 3 + for the evaluation of HER2 gene amplification status. FISH results were evaluated according to the 2013 ASCO/CAP guidelines[39]. A detailed review of all samples is provided in Supplementary Data 2.

**DNA isolation.** DNA was extracted from the primary tumours, metastases and matched normal tissue from FFPE tissue blocks after macrodissection using the QIAamp DNA FFPE Tissue Kit (Qiagen, #56404, Hilden, Germany). Double-stranded DNA (dsDNA) was quantified using the QUBIT 2.0 Fluorometer (Invitrogen) and the PicoGreen assay for double-stranded DNA. Only samples from which >1 μg of double-stranded DNA, as quantified by the two assays could be extracted, were selected for downstream molecular profiling.

**Exome and targeted sequencing analysis.** For each patient, part of the available samples was used to index the presence of SNVs using whole-exome sequencing. A total of 51 samples including at least one normal sample per patient were sequenced at a target coverage of 40 × . The putative somatic SNVs were validated by Sequenom MassARRAY in both the germline reference and cancer samples. As further validation, all available cancer samples were subjected to targeted

amplicon deep sequencing at a median coverage of $9,000 \times$ to confirm initial sequencing results and increase the accuracy of VAFs.

Whole-exome DNA libraries were generated following the manufacturer's protocol with minor modifications (Illumina TruSeq DNA library preparation kit v2). Before end-repair, a 65 °C incubation step was added to remove reversible crosslinks, after which excessive single-stranded DNA was removed enzymatically. The concentration of double-stranded DNA was assessed using the PicoGreen assay and the concentration of adapters used for ligation was adjusted accordingly. For the library enrichment, 5–7 cycles of PCR were used. Whole-exome capture was then performed using the Illumina Human Exome capture kit and libraries were sequenced on a HiSeq2000 using V3 flowcells generating $2 \times 100$ bp paired-end reads. Raw sequencing reads were mapped to the human reference genome (NCBI37/hg19) using the Burrows-Wheeler Aligner (BWA)[40] and aligned reads were processed with SAMtools[41]. Duplicate reads were removed using Picard tools. Base recalibration, local realignment around indels and SNV calling were performed using the GenomeAnalysisToolKit (GATK)[42]. Indels were called using Dindel[43]. Mutations were filtered based on mapping quality, and sequencing coverage. Somatic mutations were further filtered by comparison with the matched germline reference and using common variant databases such as the 1000 genomes project[44] and dbSNP version 132 (ref. 45).

For the targeted resequencing experiments, primers were designed using the Sequenom's MassARRAY assay design software and universal sequence tags were added manually. Amplicons encompassing the mutation of interest were generated using a first PCR (Roche FastStart High Fidelity PCR kit) with universal sequencing adapters (Access Array Barcode) containing a 10 bp index added in a second PCR. The resulting PCR products were pooled, denatured and sequenced on an Illumina MiSeq in a $2 \times 75$ bp paired-end sequencing run using a V3 flowcell. Fastq files were generated and demultiplexed using CASAVA. The raw sequencing reads were mapped to the human reference genome (NCBI37/hg19) using BWA. Read counts of both variant and reference alleles were called using GATK and manually checked in IGV[46].

**Tiering system for filtering SNVs.** A five-tier system was devised to filter SNVs for downstream analysis (Supplementary Data 3). We considered tier-0 SNVs as any SNV indexed in any sample including the matched germline reference by exome sequencing. Tier-1 SNVs are those that were further called somatic after comparison with the germline reference while tier-2 SNVs are the subset of tier-1 which have been further validated and confirmed somatic by Sequenom MassARRAY. Tier-3 SNVs are the further subset that was confirmed present by targeted deep amplicon sequencing. Due to formalin fixation and paraffin embedding, at some loci, all four bases were often observed at low frequencies. Thus, to determine the background noise of the targeted deep sequencing experiments, we selected 25 bp upstream and downstream of tier-2 SNVs. For each position, the number of reads for the reference nucleotide and the other 3 non-reference nucleotides was determined using GATK. To exclude possible SNPs or SNVs other than the one of interest, we removed all data points that showed more than 10% non-reference reads. We then pooled all the data for each mutation type (AT>CG, AT>GC, AT>TA, CG>AT, CG>GC and CG>TA) and calculated the average background signal and s.d. From these estimates, the highest value observed was 1.59%. Thus, we chose a conservative value of 3% as the final cut-off for calling a mutation present.

In the present context, several samples were matched to a given patient and we found it equally important to have a high coverage to call a mutation present in one sample, as it is to call absence in another paired sample. Thus, at the tier-3 level, we first filtered all SNVs in all matched samples to have a coverage of $>1,500 \times$ and kept samples with a minimum of 75% non-missing values. We then excluded all SNVs with $>1$ missing value across any of the matched samples previously retained. We reversed the filtering order for VAFs by first eliminating SNVs that were called absent $<3\%$ across all matched samples previously retained. Because of low CCF, it is likely that some samples have a lower total number of SNVs called present. However, a lower number of detected mutations could also be biologically grounded. Therefore, we use a lenient filtering to exclude samples with potentially low CCFs by requiring that $>20\%$ of SNVs be called present. Some samples showed a large number of SNVs or had limited starting DNA available. For those samples, we were compelled to skip validation by Sequenom MassARRAY and thus the tier-2 level. Finally, we required that for all samples where a matched-SNP array and targeted sequencing were available, that the sample displays a CCF $>30\%$ as determined by SNP array. All SNVs from that sample were then referred to as tier-4 SNVs

**SNP array analysis.** For the estimation of CNAs, a total of 64 samples were shipped for processing to the Affymetrix Research Services Laboratory. Normal and tumour DNA from FFPE samples were genotyped using the Affymetrix OncoScan FFPE Express 2.0 arrays. Routine formalin fixation and paraffin embedding frequently damages DNA causing wavy profiles, which may be wrongly interpreted as copy number aberrations. Thus, the median absolute pairwise deviation (MAPD) and the median auto-correlation (MAC) across the log₂ ratio intensities were used as quality control for the SNP arrays (Supplementary Data 2). Samples with an MAPD$>0.7$ or an MAC$>0.3$ were discarded. Samples suspected to have a low CCF on visual inspection of the BAF tracks were flagged and after

processing, those confirmed to have less than 30% CCF were discarded from downstream applications. From the remaining samples, only informative probes displaying heterozygous genotype (AB) and copy-neutral state (2) in the matched normal sample were kept for analysis. We used the added value of multiple-matched samples per patient to infer breakpoints, which may otherwise have been missed due to differences in CCF across the various samples. The log₂ ratio intensities and BAF, grouped per patient, were segmented jointly using the multitrack PCF algorithm in the R package copynumber[47] to determine common breakpoints. The penalty parameter $\gamma$ determining discontinuities in the log₂ ratio and BAF tracks was set individually for each patient after visual inspection of the segmentation profiles. Finally, all segments that were less than three s.d. away were merged with their immediate neighbours.

Integer level estimates of total copy number and major allele were obtained using GAP[48]. We compared the estimates returned by GAP with two other mainstream programs: (1) ASCAT[49] and (2) ABSOLUTE[21]. Unless otherwise stated, the parameter sets for each programme were kept at default values. For ASCAT, the parameter $\gamma$, which determines the platform-specific compression ratio, was set to 0.8. The programme returns one default estimate of ploidy and CCF, which was used in the comparison against GAP. ABSOLUTE was run in 'total copy' mode and model based evaluation against the SNP6 platform. The fraction of the genome allowed to be non-clonal was set to infinity so that the maximum number of solutions could be evaluated. ABSOLUTE returns a set of possible values for ploidy and CCF. The closest solution to that returned by GAP in Euclidean space, after rescaling ploidy values to unit distance, was chosen. The results obtained by GAP, ASCAT and ABSOLUTE are contrasted in Supplementary Fig. 10a,b. In all three cases, the Spearman's $\rho$ between the estimated CCF was high. In the case of ploidy estimates, the correlation between GAP and ASCAT was relatively low due to two reasons: (1) several matched samples from two patients displayed ploidy values outside the range considered by ASCAT and (2) at several loci displaying high copy numbers, GAP truncates the estimate to 8 while ASCAT does not thereby affecting the true estimate. We computed the Cohen's $\kappa$ coefficient between ASCAT and GAP to measure the agreement in total copy numbers and major alleles. These are displayed in Supplementary Fig. 10c while the correlation of the $\kappa$ coefficients for total copy numbers and the absolute difference in ploidy between the two algorithms is shown in panel d. The results showed that the disagreement between the two programs in the phasing of alleles and estimation of total copy numbers is in fact linked to the discordances in ploidy ($\rho = -0.898$, $P<0.01$). Because the present dataset contained matched samples and may represent a biased result, we used an additional dataset of 125 unrelated samples profiled using similar Affymetrix OncoScan SNP arrays (unpublished data) to reproduce the analysis. Supplementary Figure 11a,b shows a very good correlation between the two estimates of ploidy ($\rho = 0.885$, $P<0.01$) and CCF ($\rho = 0.907$, $P<0.01$). However, the correlation of the $\kappa$ coefficients for total copy numbers and the absolute difference in ploidy ($\rho = -0.733$, $P<0.01$) or CCF ($\rho = -0.119$, $P=0.147$) shows that it is, in fact, the incorrect estimation of global genomic mass that leads to disagreement between the two algorithms. Supplementary Figure 11f–m contrasts the results from GAP and ASCAT for four samples with increasing genomic mass and illustrates the complexity of choosing the correct solution of CCF and ploidy. Given the present context of matched samples and because GAP allowed for manual review of ploidy solutions, we opted for this package for downstream analyses taking into consideration the maximal limit of eight copies imposed by the software as follows: (1) SNVs that occurred at loci where the total copy number was 8 or a major allele count $>4$ was observed were not considered for the estimation of CCF or clonal frequency and (2) similarly for the phylogenetic reconstructions using CNAs, except in the case of high ploidy tumours (that is, 1/69, 7/67, 8/82 and 10/80) any locus displaying a total copy number of 8 or a major allele count $>4$ in any sample was removed from all matching samples of that particular patient.

**Estimation of CCF and clonal frequencies from tier-4 SNVs.** The CCF was estimated both individually from each tier-4 SNV and globally from the whole set of tier-4 SNVs in samples where a matched-SNP array was available. Let $q_t$ denote the total copy number at the mutated locus, $q_1$ denote the minor copy number and $q_2$ denote the major copy number such that $q_2 \geq q_1$, $q_t = q_1 + q_2$ and $q_1$, $q_2$ and $q_t \in \alpha$. Let $s_q$ denote the number of mutated copies such that $s_q \in \{1, \ldots, q_2\}$. Let $f_{sq}$ denotes the expected VAF of the SNV where $f_{sq} \in [0,1]$. Then, $f_{sq}$ is related to $s_q$ and $\alpha$, the CCF, as follows:

$$f_{sq} = s_q \left( \frac{\alpha}{\alpha q_t + 2(1-\alpha)} \right) \qquad (1)$$

where $\alpha$ is the variable that we are trying to estimate while $\hat{f}$ is taken to be the observed VAF. We denote the estimate of CCF from sequencing as $\hat{\alpha}$. The above equation can be rearranged such that

$$\hat{\alpha} = \frac{2\hat{f}}{s_q - \hat{f}(q_t - 2)} \qquad (2)$$

Let $n$ be the total number of sequencing reads that cover the mutated locus. Then

$$\Pr\left(X - n\hat{f}\right) = \omega_{sq} \text{Beta}\left(f_{sq} \middle| n\hat{f} + 1, n(1-\hat{f}) + 1\right) \qquad (3)$$

where $X$ is the number of mutated reads for a given SNV, $w_{sq}$ specify the mixture weights for each possible value of $s_q$. Computation of $f_{sq}$ requires prior knowledge of $\alpha$. To estimate $s_q$ and for individual SNVs irrespective of other mutations, we make the assumption that $\alpha$ is known. Thus, $q_t$, $q_1$, $q_2$ and $\alpha$ are plugin quantities obtained from the corresponding SNP array. Then

$$s_q^* = \underset{s_q \in \{1,\dots,q_2\}}{\operatorname{argmax}} \left\{ \Pr(X = n\hat{f}) \right\} \qquad (4)$$

and the sequencing error $e \in [0,1)$ is modelled after Purdom et al.[50] such that

$$f_e = \frac{1}{3-2e}\left((3-4e)f_{sq} + e\right) \qquad (5)$$

replaces $f_{sq}$ in the Beta distribution. We use uniform priors over the range of possible values of $s_q$ and $e = 0.01$. These form the basis of the pointwise estimates of CCFs (Supplementary Fig. 10b). To relax the requirement on prior knowledge of $\alpha$, we define the likelihood function over all tier-4 SNVs present in a given sample as

$$\mathcal{L}(f|n, s_q, \omega_{sq}) = \sum_{s_q \in \{1,\dots,q_2\}} \Pr\left(X - n\hat{f}\right) \qquad (6)$$

At a given value of $\alpha$, we compute the log of $L$ and iteratively adjust the weights $w_{sq}$ until $L$ converges or a maximum of 100 iterations is reached. The global CCF, $\alpha^*$, is the value that maximizes $L$ such that

$$\alpha^* = \underset{\alpha \in \{0.1,\dots,0.9\}}{\operatorname{argmax}} \left\{ \mathcal{L}(\hat{f}|\alpha) \right\} \qquad (7)$$

An example is shown in Supplementary Fig. 10e–f and $\alpha^*$ is compared to the estimate of GAP in Supplementary Fig. 10g. The CCF and the clonal frequency of SNVs are related and the latter was computed jointly for all samples belonging to a given patient using PyClone[51] from the estimates of major and minor copy numbers returned by GAP. We used a Beta Binomial distribution with parental copy number option and default parameter settings except for the sequencing error which was set to 0.01 and the tumour content which was set to the global CCF, $\alpha^*$, estimated above.

**Phylogenetic analysis of SNVs.** The raw VAFs of tier-3 SNVs from targeted resequencing were converted into binary calls based on a threshold of 3%. We initially intended to filter in only fully clonal tier-3 SNVs using this conservative cut-off and infer the phylogeny for individual patients using the Dollo parsimony method and a branch and bound exhaustive search for the best phylogenetic reconstruction as described in Felsenstein[52] using the programme PHYLIP. The outgroup used for rooting the phylogenies was one where all the characters were set to the ancestral state 0. The Dollo parsimony criterion minimizes homoplasies at the expense of reversions in later branches and the criteria for determining the best phylogenetic tree is minimizing the number of such reversions. Despite this, several phylogenetic trees can be equally parsimonious. Instead of collapsing the trees using consensus methods, we used the corresponding CNA based tree to break ties and infer the correct phylogeny. The trees in Newick format were rendered using the R package ape[53]. The heat maps representing the tier-3 SNVs were ordered according to the topology and branch lengths of their corresponding phylogenetic trees via an ancestral state reconstruction using the accelerated transition model[54,55] as provided in the R package phangorn[56]. The phylogenetic trees for each patient are shown in Supplementary Fig. 7. Each predicted reversion of SNV was manually verified against the underlying CNA profile. The tier-3 level does not take into account the CCF of the samples. It is possible that, despite the previous filters, very low CCF samples with overall fewer positive mutation calls are included. These samples would lead to early branches in the trunk of the phylogenies. Thus, we reproduced the same analysis with samples having tier-4 SNVs. The results are shown in Supplementary Fig. 8.

**Phylogenetic analysis of CNAs.** The major and minor copy numbers returned by GAP were modelled using a transducer-based pairwise comparison function using the programme MEDICC[57]. For near-diploid samples, we assume a pure diploid outgroup with no copy number aberrations that is, 2/1 (total copy number/major allele) to root the phylogenies. In the case of tetraploid samples, we included an additional step to phase CNAs relative to the whole-genome duplication event within the phylogeny of the given patient. We first used the classic approach to infer an intermediate tree with correct topology irrespective of branch lengths. For the major or minor copy and at each locus, we compute a parsimony score, which is the sum of branch lengths of the intermediate tree rooted using any of the four tetraploid ancestral states 6/4, 4/2, 4/4 and 2/2 (total copy number/major allele). These represent the copy number states 3/2, 2/1, 2/2 and 1/1 following a whole-genome duplication event. We used all possible permutations of observed copy numbers at the internal nodes except for $0 \rightarrow 1$ transitions. We chose the intermediate tree and thus the related tetraploid ancestral state obtaining the minimum score. Ties, if present, are broken by summing the intermediate tree length with the CNAs occurring prior to the whole-genome duplication that is, 1, 0, 2 and 1, respectively. Finally, the global phylogenetic tree is inferred using the classic approach jointly at all loci and rooted using the tetraploid ancestor as

outgroup. The phylogenetic trees are shown in Figs 3–5 of the main text and Supplementary Fig. 9. Support values for the phylogenetic trees were obtained by resampling the pairwise distance matrix 100 times with added Gaussian noise and counting similar bipartitions between the resulting trees and the original phylogeny.

**Data availability.** The sequencing and SNP array data have been deposited at the European Genome-Phenome Archive (http://www.ebi.ac.uk/ega/), which is hosted by the European Bioinformatics Institute, under accession number EGAS00001000760.

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

## Acknowledgements

We would like to extend our gratitude to the families of the deceased patients who participated in this study. The authors would further like to thank N. Kheddoumi, E. Azumah, M. Pekár and E. Samodai for technical assistance, O. Kiss, I. Illyés, L. Madaras, A. Kovács, G. Lotz, Z. Baranyai, B. Járay, T. Glasz, T. Herbert, K. Simon, C. Diczházi, I. Kaszás, G. Lukács Tóth, T. Barna, F. Salamon and G. Bodoky for collection of study material, Z. Schaff, J. Tímár and their colleagues at the 2nd Department of Pathology, Semmelweis University for supporting this research programme. D.B. and C.S. are supported by the Belgian Fonds National de la Recherche Scientifique (F.R.S-FNRS). C.D. is supported by a grant from the Brussels Region—Impulse Programme Life Sciences and by Les Amis de l'Institut Bordet. B.S. is supported by the Susan G. Komen and American Joint (JDC) and A.M.S. by the European Union and the State of Hungary, co-financed by the European Social Fund in the framework of TÁMOP 4.2.4.A/2-11-1-2012-0001 National Excellence Program. This study was supported by grants from the MEDIC Foundation, Les Amis de l'Institut Bordet, MKOT-Roche-2012, TÁMOP 4.2.2/B-10/1-2010-0013 and TÁMOP 4.2.1.B-09/1/KMR-2010-0001.

## Author contributions

C.D. and C.S. conceived the study. B.S., A.M.S., Z.I.N., Z.F., A.-M.T, M.D., G.S. and J.K. provided the clinical specimens. B.S., A.M.S., Z.I.N., Z.F., A.-M.T., D.L. and J.K. performed the histopathological assessment of the samples. P.-Y.A. and G.R. processed the samples. D.S. aligned the sequencing reads and called the mutations. D.B. analysed the SNP arrays, reconstructed the phylogenies and rendered the artwork. D.B., D.S. and B.S. reviewed and assembled all the results. D.B., D.S., B.S., G.Z., L.P., C.S., M.P., D.L., J.K., D.L. and C.D. interpreted the results. D.B., D.L., C.S. and C.D. wrote the manuscript. All the authors read and approved the final manuscript.

## Additional information

**Competing interests:** The authors declare no competing financial interests.

