## [Peer Review File · Nature Communications]

Reviewers' comments:

Reviewer #1 (Remarks to the Author):

Herein, the author's have inferred phylogenetic trajectories in ten breast cancer patients, based on both single nucleotide variants (SNV) and copy number alterations (CNA). Two types of metastatic dissemination are observed: (1) monoclonal seeding of disseminated tumor cells, followed by metastasis-to-metastasis seeding and (2) multiple seeding events from the primary tumor (diffusely metastatic), alongside daughter metastasis-to-metastasis seeding. The former was observed in 8/10 patients with early stage breast cancer, the later in 2/10 patients with advanced stage disease.

1. Supplemental figure 1 - perhaps wrong figure uploaded? The figure itself shows ancestral state reconstruction and associated phylogenetic trees for three patients, however the legend states that this figure should show power calculations for patient 8/82. Appears to be a discrepancy...

2. Style: Overall, language could be improved, especially in the introduction and discussion. Additionally, in the discussion, the authors do not need to justify the sample size. It is necessary to provide this justification to the reviewers and editor, but detracts from communicating the actual science and discovery here. Especially the second to last paragraph, which is entirely dedicated to explaining why the author's did not exclude patients based on clinical subtype. I think this paragraph is entirely unnecessary.

3. Discussion Paragraph 2 on the rationale for primary surgery in patients with de novo metastasis is confusing and not clearly reflecting the strong clinical data showing its lack of clinical benefit. Primary surgery is not really entertained as a treatment option for the vast majority of patients and the few prospective studies that have looked at this in a meaningful way show no benefit from primary surgery. Only retrospective studies have ever suggested any benefit from primary surgery, and these are strongly biased by attending selection. Instead, the authors might discuss how the observation of metastasis-to-metastasis seeding might support a clinical trial to evaluate whether resection of oligo-metastasis improves survival. Currently, there is no clinical data evaluating the impact of resecting oligo-mets on survival.

4. The author's response to the initial point #4 is not satisfactory. The figures are not too complex to add support values reflecting how well nodes are supported in the model used to generate the phylogenetic tree. In fact, in the initial MEDICC publication, which the authors use and cite as superior, approximate support values are indicated on all trees to indicate how often each split was observed in trees reconstructed after simple bootstrapping (resampling the distance matrix with added Gaussian noise). The authors should further test their distance distribution for the molecular clock hypothesis, as described in MEDICCquant; as this can also significantly increase reconstruction accuracy.

5. The authors should include the secondary suggestion about scaling node with subclonal frequency, or at least make all tree nodes the same size or explaining the meaning of the different node sizes.

6. Doubts about the CNA based phylogenetic tree for patient 1/69, which places M5 as a very early branching metastasis, would be alleviated by support values reflecting confidence in phylogenetic tree construction.

7. It would be interesting to discuss how the amplitude of the normalized phylogenetic distance

(Figure 6A-B) is much higher for CNA as compared to SNV, and therefore allows for higher resolution when mapping subclonal architecture...

Reviewer #2 (Remarks to the Author):

The authors have responded to each of the reviewers' specific comments, performed several additional analyses, and significantly revised the manuscript. The revised manuscript is improved. Novelty and sample size are still an issue, but the findings are certainly worth reporting.

Reviewer #3 (Remarks to the Author):

The authors addressed most of the Reviewers' comments adequately for the most part in this much improved manuscript. The paper would still be significantly strengthened with more cases.

Reviewers' comments:

Reviewer #1 (Remarks to the Author):

Herein, the author's have inferred phylogenetic trajectories in ten breast cancer patients, based on both single nucleotide variants (SNV) and copy number alterations (CNA). Two types of metastatic dissemination are observed: (1) monoclonal seeding of disseminated tumor cells, followed by metastasis-to-metastasis seeding and (2) multiple seeding events from the primary tumor (diffusely metastatic), alongside daughter metastasis-to-metastasis seeding. The former was observed in 8/10 patients with early stage breast cancer, the later in 2/10 patients with advanced stage disease.

1. Supplemental figure 1 - perhaps wrong figure uploaded? The figure itself shows ancestral state reconstruction and associated phylogenetic trees for three patients, however the legend states that this figure should show power calculations for patient 8/82. Appears to be a discrepancy....

There is, in fact, a discrepancy between the figure displayed and the legend. The Reviewer will find the correct figure and legend which should appear as Supplementary Figure 1 of the manuscript as Figure 1 at the end this point-by-point answer.

2. Style: Overall, language could be improved, especially in the introduction and discussion. Additionally, in the discussion, the authors do not need to justify the sample size. It is necessary to provide this justification to the reviewers and editor, but detracts from communicating the actual science and discovery here. Especially the second to last paragraph, which is entirely dedicated to explaining why the author's did not exclude patients based on clinical subtype. I think this paragraph is entirely unnecessary.

The authors take due note of the Reviewer's comment. The authors have made revisions to the introduction and discussion. Additionally, the paragraph detailing why patients were not excluded has been removed.

3. Discussion Paragraph 2 on the rationale for primary surgery in patients with *de novo* metastasis is confusing and not clearly reflecting the strong clinical data showing its lack of clinical benefit. Primary surgery is not really entertained as a treatment option for the vast majority of patients and the few prospective studies that have looked at this in a meaningful way show no benefit from primary surgery. Only retrospective studies have ever suggested any benefit from primary surgery, and these are strongly biased by attending selection. Instead, the authors might discuss how the observation of metastasis-to-metastasis seeding might support a clinical trial to evaluate whether resection of oligo-metastasis improves survival. Currently, there is no clinical data evaluating the impact of resecting oligo-mets on survival.

Indeed, the authors concur fully with the Reviewer as far as primary surgery in *de novo* metastatic patients is concerned. Our data suggest that tumor resection could probably limit metastatic dissemination from the primary tumor in advanced stage breast cancer. However, in absence of the primary tumor, metastasis-to-metastasis disseminations may

still occur. There is no consensus about surgery at the primary site in the metastatic setting as the literature does not show consistent data of improvement in patient survival:

- a) Preclinical studies from animal models have shown that removing the primary tumor might cause an increase in the metastatic spread (Fisher, Gunduz et al. 1989 & Demicheli et al. 1997).
- b) Numerous retrospective reviews found improved survival in patients who underwent surgery for the primary tumor (Rapiti et al. 2006, Babiera, et al. 2006, Gnerlich et al. 2007 & Neuman et al. 2010) and similarly, a recent meta-analysis reported fewer competing medical comorbidities and lower metastatic burden in the surgery group (Harris et al. 2013). However, as subsequently highlighted, these results were likely tainted by a selection bias in patients.
- c) To the best of our knowledge, only one prospective, randomized, controlled clinical trial has been published (Badwe et al. 2015) and quite disconcertingly, the authors did not find any survival benefit to loco-regional management. Similarly, another prospective randomized study presented in abstract form did not find any benefit in overall survival, although a trend towards survival benefit was observed in the subgroup of patients with ER-positive disease and distant metastases limited to the bone (Soran et al. 2013).

Due to the contradictory findings outlined above, the concept of resecting oligo-metastases in early stage patients who underwent surgery and relapsed at a distant organ years after the initial procedure in order to prolong survival is also very attractive. However, the same argument as primary resection in *de novo* metastatic patients could be leveraged against this. Namely, that making a decision on surgery is always a complex issue that has to be dealt with on a case-by-case basis taking into account the accessibility of the organ for surgical intervention, the extensiveness of the disease and other personal factors relevant to the individual patient status. Thus, the authors make a brief comment on both in the manuscript but remain conservative by concluding that to have a better understanding of the problem, more prospective randomized clinical trials are needed.

4. The author's response to the initial point #4 is not satisfactory. The figures are not too complex to add support values reflecting how well nodes are supported in the model used to generate the phylogenetic tree. In fact, in the initial MEDICC publication, which the authors use and cite as superior, approximate support values are indicated on all trees to indicate how often each split was observed in trees reconstructed after simple bootstrapping (resampling the distance matrix with added Gaussian noise). The authors should further test their distance distribution for the molecular clock hypothesis, as described in MEDICCquant; as this can also significantly increase reconstruction accuracy.

The authors agree with the Reviewer at part (i) of his/her comment. Support values for each split in the phylogenetic trees have been added to Supplementary Figure 9 of the manuscript. For the Reviewer's convenience, this has been added as Figure 2 of this point-by-point answer. In addition, the authors found it useful to compute a global P-value for each phylogenetic. As before, this was computed by resampling the pairwise distance matrix of copy number aberrations 100X with added noise i.e. a random number generated from a Normal distribution with mean equal to the distance between two samples and variance

equal to the square root of that distance. The 100 replicate trees were then compared to the initial phylogenetic tree. However, instead of counting similar bipartitions, the Robinson-Foulds distance was computed as a metric of tree likeness. We then exhaustively generated all possible phylogenetic trees with $n + 1$ leaves for each patient, where n is the number of samples. As before, we computed the Robinson-Foulds distance between these phylogenetic trees and the initial tree built using MEDICC to obtain a null distribution of Robinson-Foulds distances. The two distributions i.e. by distance matrix resampling using MEDICCquant and by exhaustive generation of all trees with $n + 1$ leaves, were compared using a one-sided two-sample Kolmogorov-Smirnov test for empirical cumulative distribution functions. Table 1 of this point-by-point answer lists the P-values obtained.

Pertaining to part (ii) of the Reviewer's comment about the molecular clock hypothesis, the authors would like to highlight the following. If applied to the present case, the latter hypothesis would entail that copy number aberrations and point mutations accumulate at a constant rate in the different metastases. If this holds true, it would effectively mean that the phylogenetic trees are ultrametric i.e. the sum of branch lengths from the root to any branch tip are equal for all metastases. This property of phylogenetic trees should not be confused with any measure of reconstruction accuracy. However, the authors agree with the Reviewer that the story wouldn't be complete without exploring this aspect. MEDICCquant tests this hypothesis of a constant evolutionary rate along all branches by summing over all leaves, the squared difference between a leaf node and its diploid root divided by the square root of the given distance. Because of possible measurement errors, this number is not a perfect multiple of n and thus, the null distribution is a Chi-squared with $n - 1$ degrees of freedom. The P-values obtained from our data are presented in Table 2 of this point-by-point answer. Our results show that only one phylogenetic tree, that of patient 5/87, is ultrametric and supports a constant evolutionary rate along all branches whilst the remaining phylogenies show a strong tendency to the opposite. Whilst this might at first glance appear surprising, it is not necessarily so. Indeed, patterns of copy number aberrations in cancers are very often complex. For instance, the phenomenon of whole genome duplication has been known for a while and its effect on accelerating chromosomal instability and evolution of the cancer genome is only now being appreciated [10]. Similarly, the recent publication of Gao *et al.* [11] who found that copy number aberrations are acquired in short punctuated bursts rather than gradually further lends support, if need be, to our results. Lastly, different metastases within the same patient are harboured at different organ sites e.g. bone metastasis versus liver metastasis and likely to have different rates of cell division thereby invalidating the assumption of a constant evolutionary rate along all branches.

5. The authors should include the secondary suggestion about scaling node with subclonal frequency, or at least make all tree nodes the same size or explaining the meaning of the different node sizes.

The main figures and the supplementary figures have been updated. All the nodes are now of the equal size.

6. Doubts about the CNA based phylogenetic tree for patient 1/69, which places M5 as a very early branching metastasis, would be alleviated by support values reflecting confidence in phylogenetic tree construction.

Figure 2 of this point-by-point answer show that the first split of the phylogenetic tree of patient 1/69 was found in 100% of all trees generated by resampling with added Gaussian noise using MEDICCquant.

7. It would be interesting to discuss how the amplitude of the normalized phylogenetic distance (Figure 6A-B) is much higher for CNA as compared to SNV, and therefore allows for higher resolution when mapping subclonal architecture...

The methodological aspect of inferring the subclonal architecture from a bulk tumor tissue has received considerable attention in the literature [12-14]. These early modelling frameworks have also been modified to account for multiple related samples using phylogenetic trees [15]. The authors believe that the answer to the Reviewer's question should be nuanced as it is more a matter of practicality than personal liking. For instance, in Figure 4 of the manuscript and Figure 1.4 of the initial point-by-point answer, we showed that there was a horizontal transfer between the ovarian metastasis (M3) and the adrenal gland metastasis (M2) of patient 2/57. This event was only detectable using point mutations due to the subtle cellular composition of the sample and high accuracy of variant allele fractions at high sequencing depth as evidenced by Figure 1.5 of the initial point-by-point answer. This is not an isolated case as exemplified by Figure 1.9 of the initial point-by-point answer detailing the case of patient 1/69. Yet other examples could be cited as for instance, the work of Gundem *et al.* [16]. However, our data also suggest that, given the correlation with time, for patients who have a long overall survival, the extent of copy number aberrations private to or shared among a group of metastases is likely to be comparatively larger than in *de novo* metastatic patients. Thus, the authors are of the opinion that at the lower end of patient overall survival, both substitutions/indels and copy number aberrations are required whilst at the other extreme, copy number aberrations are equally good as point mutations for the purpose of phylogenetic reconstruction.

Supplementary Fig. 1

Figure 1: Power calculations for samples of patient 8/82. (a) Required depth of sequencing coverage as a function of CCF (%) and CNA in the worst-case scenario of one mutated copy (1^*) to reach a statistical power of 95% given a sequencing error rate $e = 1.59E-2$ and a FPR = $5E-7$ to index a fully clonal SNV. The black vertical lines indicate the CCF of samples. (b) Same as a function of subclonal cell fraction (%).

Figure 2: Phylogenetic trees with support values at each split.

Table 1: Global P-values obtained using exhaustive enumeration

Patient	P-value
1/69	9.719E-07
2/57	1.829E-05
3/92	3.307E-39
4/71*	-
5/87	3.085E-04
6/91*	-
7/67	9.219E-06
8/82	2.188E-20
9/68	9.219E-06
10/80	7.997E-23

* The phylogenetic trees contain $n = 2$ samples. Thus, the bootstrap values and the global P-value are irrelevant

Table 2: P-values for test of molecular clock hypothesis

Patient	P-value
1/69	0.995
2/57	0.889
3/92	0.551
4/71	0.952
5/87	0.048
6/91	0.916
7/67	0.970
8/82	0.990
9/68	0.832
10/80	0.327

References

1. Fisher B, Gunduz N, Coyle J, Rudock C, Saffer E. Presence of a growth-stimulating factor in serum following primary tumor removal in mice. *Cancer Res.* 1989 Apr 15;49(8):1996-2001.
2. Demicheli R, Retsky MW, Swartzendruber DE, Bonadonna G. Proposal for a new model of breast cancer metastatic development. *Ann Oncol.* 1997 Nov;8(11):1075-80.
3. Rapiti E, Verkooijen HM, Vlastos G, Fioretta G, Neyroud-Caspar I, Sappino AP, Chappuis PO, Bouchardy C. Complete excision of primary breast tumor improves survival of patients with metastatic breast cancer at diagnosis. *J Clin Oncol.* 2006 Jun 20;24(18):2743-9. Epub 2006 May 15.
4. Babiera GV, Rao R, Feng L, Meric-Bernstam F, Kuerer HM, Singletary SE, Hunt KK, Ross MI, Gwyn KM, Feig BW, Ames FC, Hortobagyi GN. Effect of primary tumor extirpation in breast cancer patients who present with stage IV disease and an intact primary tumor. *Ann Surg Oncol.* 2006 Jun;13(6):776-82. Epub 2006 Apr 17.
5. Gnerlich J, Jeffe DB, Deshpande AD, Beers C, Zander C, Margenthaler JA. Surgical removal of the primary tumor increases overall survival in patients with metastatic breast cancer: analysis of the 1988-2003 SEER data. *Ann Surg Oncol.* 2007 Aug;14(8):2187-94. Epub 2007 May 24.
6. Neuman HB, Morrogh M, Gonen M, Van Zee KJ, Morrow M, King TA. Stage IV breast cancer in the era of targeted therapy: does surgery of the primary tumor matter? *Cancer.* 2010 Mar 1;116(5):1226-33. doi: 10.1002/cncr.24873.
7. Harris E, Barry M, Kell MR. Meta-analysis to determine if surgical resection of the primary tumour in the setting of stage IV breast cancer impacts on survival. *Ann Surg Oncol.* 2013 Sep;20(9):2828-34. doi: 10.1245/s10434-013-2998-2. Epub 2013 May 8.
8. Badwe R, Hawaldar R, Nair N, Kaushik R, Parmar V, Siddique S, Budrukkar A, Mitra I, Gupta S. Locoregional treatment versus no treatment of the primary tumour in metastatic breast cancer: an open-label randomised controlled trial. *Lancet Oncol.* 2015 Oct;16(13):1380-8. doi: 10.1016/S1470-2045(15)00135-7. Epub 2015 Sep 9.
9. Soran A, Ozmen V, Ozbas S, Karanlik H, Muslumanoglu M, Igci A, Canturk Z, Utkan Z, Ozaslan C, Evrensel T, Uras C, Aksaz E, Soyder A, Ugurlu U, Col C, Cabioglu N, Bozkurt B, Dagoglu T, Uzunkoy A, Dulger M, Koksall N, Cengiz O, Gulluoglu B, Unal B, Atalay C, Yildirim E, Erdem E, Salimoglu S, Sezer A, Koyuncu A, Gurleyik G, Alagol H, Ulufi N, Berberoglu U, Kennard E, Kelsey S, Lembersky B. Abstract S2-03: Early follow up of a randomized trial evaluating resection of the primary breast tumor in women presenting with de novo stage IV breast cancer; Turkish study (protocol MF07-01). *Cancer Research.* 2013; 73(24):S2-3. 10.1158/0008-5472.SABCS13-S2-03.
10. Dewhurst SM, McGranahan N, Burrell RA, Rowan AJ, Grönroos E, Endesfelder D, Joshi T, Mouradov D, Gibbs P, Ward RL, Hawkins NJ, Szallasi Z, Sieber OM, Swanton C. Tolerance of whole-genome doubling propagates chromosomal instability and accelerates cancer genome evolution. *Cancer Discov.* 2014 Feb;4(2):175-85. doi: 10.1158/2159-8290.CD-13-0285.
11. Gao R, Davis A, McDonald TO, Sei E, Shi X, Wang Y, Tsai PC, Casasent A, Waters J, Zhang H, Meric-Bernstam F, Michor F, Navin NE. Punctuated copy number evolution and clonal stasis in triple-negative breast cancer. *Nat Genet.* 2016 Oct;48(10):1119-30. doi: 10.1038/ng.3641.
12. Carter SL, Cibulskis K, Helman E, McKenna A, Shen H, Zack T, Laird PW, Onofrio RC, Winckler W, Weir BA, Beroukhim R, Pellman D, Levine DA, Lander ES, Meyerson M,

- Getz G. Absolute quantification of somatic DNA alterations in human cancer. *Nat Biotechnol.* 2012 May;30(5):413-21. doi: 10.1038/nbt.2203.
13. Roth A, Khattra J, Yap D, Wan A, Laks E, Biele J, Ha G, Aparicio S, Bouchard-Côté A, Shah SP. PyClone: statistical inference of clonal population structure in cancer. *Nat Methods.* 2014 Apr;11(4):396-8. doi: 10.1038/nmeth.2883. Epub 2014 Mar 16.
 14. Oesper L, Mahmoody A, Raphael BJ. THetA: inferring intra-tumor heterogeneity from high-throughput DNA sequencing data. *Genome Biol.* 2013 Jul 29;14(7):R80. doi: 10.1186/gb-2013-14-7-r80.
 15. El-Kebir M, Oesper L, Acheson-Field H, Raphael BJ. Reconstruction of clonal trees and tumor composition from multi-sample sequencing data. *Bioinformatics.* 2015 Jun 15;31(12):i62-70. doi: 10.1093/bioinformatics/btv261.
 16. Gundem G, Van Loo P, Kremeyer B, Alexandrov LB, Tubio JM, Papaemmanuil E, Brewer DS, Kallio HM, Högnäs G, Annala M, Kivinummi K, Goody V, Latimer C, O'Meara S, Dawson KJ, Isaacs W, Emmert-Buck MR, Nykter M, Foster C, Kote-Jarai Z, Easton D, Whitaker HC; ICGC Prostate UK Group, Neal DE, Cooper CS, Eeles RA, Visakorpi T, Campbell PJ, McDermott U, Wedge DC, Bova GS. The evolutionary history of lethal metastatic prostate cancer. *Nature.* 2015 Apr 16;520(7547):353-7. doi: 10.1038/nature14347. Epub 2015 Apr 1.